# Effect of Different Amounts of Hybrid Barley in Diets on the Growth Performance and Selected Biochemical Parameters of Blood Serum Characterizing Health Status in Fattening Pigs

**DOI:** 10.3390/ani10111987

**Published:** 2020-10-29

**Authors:** Anna Szuba-Trznadel, Tomasz Hikawczuk, Małgorzata Korzeniowska, Bogusław Fuchs

**Affiliations:** 1Department of Animal Nutrition and Feed Management, The Faculty of Biology and Animal Sciences, Wrocław University of Environmental and Life Sciences, J. Chełmońskiego 38 C, 51-630 Wrocław, Poland; tomasz.hikawczuk@gmail.com (T.H.); boguslaw.fuchs@upwr.edu.pl (B.F.); 2Department of Functional Food Products Development, The Faculty of Biotechnology and Food Sciences, Wrocław University of Environmental and Life Sciences, Chełmońskiego 37, 51-630 Wrocław, Poland; malgorzata.korzeniowska@upwr.edu.pl

**Keywords:** fattening pigs, performance, biochemical parameters, health status, hybrid barley

## Abstract

**Simple Summary:**

The aim of the present study was to determine the effect of dietary hybrid barley and/or wheat on production parameters, selected biochemical parameters of blood serum characterizing health status in fattening pigs. The use of hybrid barley as the basic ingredient of diets for fattening pigs provided similar production parameters as those obtained with wheat. No significant differences were noted in case of performance results and meatiness of fatteners. However, usage of hybrid barley with high level in diet decreased level of total cholesterol and LDL (low-density lipoprotein fraction) fraction in blood. It means that barley had a beneficial effect on blood lipid indices.

**Abstract:**

The aim of the study was to determine the effect of dietary hybrid barley and/or wheat on production parameters, selected biochemical parameters of blood serum characterizing health status in fattening pigs. In group I, hybrid barley constituted 80% of feed; in II—wheat and hybrid barley were used, each in amount of 40% feed; in III—contained 80% of wheat. No significant differences were noted in case of performance results (body weight gains, feed intake, and feed conversion ratio) and meatiness of fatteners. All estimated biochemical indices determined in serum were within normal range. Usage of 80% hybrid barley decreased concentration of total cholesterol, low-density lipoprotein fraction (LDL), and triglycerides in blood (*p* < 0.05). However, high-density lipoprotein fraction (HDL) content increased (*p* < 0.01) up to 1.04 mmol·dm^−3^, comparing to the group with 80% of wheat (0.84 mmol·dm^−3^). Summarized, the diet with high level of barley had a beneficial effect on blood lipid indices, what indicate a good health status of all animals.

## 1. Introduction

Barley is a basic and, to a large extent, indispensable ingredient of feeds for fattening pigs. A rise in wider barley use in stock feeding is most of all consequential to improvement of its yields. Breeding works have led to development of hybrid barley characterized by higher and more stable yields under different cultivation and environmental conditions and by better technological quality parameters. Field observations show that under good weather conditions, the yield of hybrid barley is even 14% higher compared to conventional varieties. Moreover, hybrid varieties are characterized by a higher content of crude protein. They also have only slightly lower energy value than wheat because hybrid barley contains less crude fiber than conventional varieties (www.syngenta.pl). Therefore, development of new swine breeds, as well as changes in preferences of the meat processing industry, prompted feed science to evaluate hybrid grains in feeding of fattening pigs.

Research has shown that barley has a beneficial effect on the quality of pork as a raw material for further processing by the meat industry [1,2]. Barley boosts meat and backfat palatability, consistency, and stability [3,4,5]. Moreover, dietary fiber present in barley grain, especially β-glucans, effectively decreases level of total cholesterol and LDL-cholesterol in animals but also in humans [6,7]. Therefore, barley could be used as a substitute for another grain in the diets of growing and fattening pigs, without compromising muscle quality or palatability. However, there is a scarcity of research on the application of hybrid barley in feeding modern high meat swine, and in particular, there is notable insufficiency of data regarding growth performance parameters and its effect on cholesterol and lipids fraction and also proteinogram in blood. The above reports prompted the authors to carry out a study aimed at checking whether hybrid barley used to feed fattening pigs influences production performance and selected biochemical blood serum parameters. In the present study, in the fattening process pigs were fed complete diets with high and medium contents of hybrid barley. Wheat grain was used as a reference because due to its properties it is used for the production of very high-nutritional value feed mixtures. As well as because the cost of harvesting hybrid barley is similar to that of wheat.

The aim of the present study was to determine the effect of dietary hybrid barley and/or wheat on production parameters and selected biochemical blood serum parameters in fattening pigs.

## 2. Materials and Methods

### 2.1. Animals, Diets, and Feeding

The experiment was approved by the Local Ethical Review Committee for Animal Experiments in Wrocław, Poland (protocol no. 002/2019). The nutritional experiment was carried out at a private pig farm. The study was conducted on 144 fattening pigs (Polish large white × Polish landrace crossbreds), trying to keep sex ratio 1:1. Fattening lasted 78 days, from average body weight of 55 to 120 kg. During the experiments, the animals were kept in group pens (eight animals each) equipped with nipple drinkers. A semi ad libitum feeding system was used (the amount of the mixture given to pigs was increased depending on the individual intake of the daily feed dose).

Hybrid varieties of barley Hyvido^TM^ (Syngenta Co., Warszawa, Poland) were used in the study. The animals were randomly assigned to three experimental groups of equal size differing in the type of complete diet used for feeding that was prepared based on hybrid barley and/or wheat. In group I, the diet contained 80% of hybrid barley. Group II was given an equal amount of wheat and hybrid barley (40% each). The diet for group III contained 80% wheat. The experimental design is illustrated in Table 1.

Before preparation of the experimental diets, chemical analysis of hybrid barley and wheat was performed in order to determine their nutritional value and amino acid index. The energy value of the cereals was calculated based on digestibility coefficients listed in the Nutrient Requirements for Swine [8]. Energy value of barley was calculated at the level of 12.9 MJ ME, while of wheat at 13.5 MJ ME. The chemical composition and nutritional value of the studied cereals are presented in Table 2. The contents of basic nutrients in cereal grains and diets were determined according to Official Methods of Analysis of AOAC International (AOAC) [9] (dry matter (DM, AOAC: 934.01); crude protein (CP, Kjeldahl method, AOAC: 984.13); crude ash (CA, AOAC: 942.05); ether extract (EE, Soxhlet method, AOAC: 920.39A) with the use of a BUCHI Extraction System B-811 (BÜCHI, Flawil, Switzerland); crude fiber (CF, Henneberg and Stohmann method, AOAC: 978.10). The amino acid (AA) profile was estimated by ion-exchange chromatography using an Amino Acids Analyzer AAA 400 (INGOS, Prague, the Czech Republic) according to standard protocol AOAC [9] (AOAC: 994.12). Tryptophan was determined using a spectrophotometer 2000 RS (Aqualytic, Dortmund, German) at a wavelength of 590.0 nm—AOAC [9] (AOAC: 988.15). The prepared diets had a free-flowing form. The composition and nutritional value of the diets is presented in Table 3.

The following production data were collected during the course of the experiment: body weight (BW) of fatteners (at the beginning and the end of the experiment); feed intake (FI) and animals’ losses. Then average daily gains (ADG) and feed conversion ratio (FCR) were calculated.

The lean meat content in the pig carcass was measured on the slaughter line by the IM-03 apparatus as a part of the post-slaughter classification of pork. The measurement was taken at one point: between the 3rd and 4th ribs. The meatiness was determined as the data of fat and muscle thickness. The slaughter yield was calculated as a percentage of meat content in the carcass to total BW.

### 2.2. Measurement of Blood

On the last day of the fattening period, blood was collected from the external jugular vein (vena jugularis externa) directly into the Sarstedt type-tubes embedded with a clotting activator. Following, the samples were centrifuged using the MPW-223e laboratory centrifuge (MPW, Poland). The blood samples were collected from 10 randomly selected animals from each group in order to determine the contents of total cholesterol, high- (HDL) and low-density lipoprotein (LDL) fractions, and triglycerides. ABX PENTRA Cholesterol test (Horiba ABX SAS, France) based on an enzymatic photometric Trinder’s reaction was used for quantitative diagnostic determination of cholesterol (CP reagent), high density lipoprotein cholesterol (HDL Direct CP reagent), and low density lipoprotein cholesterol (LDL Direct CP reagent). ABX PENTRA Triglycerides test (CP reagent) was used for quantitative determination of triglycerides based on an enzymatic colorimetric assay. In addition, biochemical parameters characterizing animal health status were analyzed. Serum proteinogram (total protein, albumins) was obtained according to standard methods. The BCA (bicinchoninic acid) Protein Assay Kit (Sigma-Aldrich, St Louis, MO, USA) was used for quantitation of total protein (TP), and its fraction using the filter paper electrophoresis. Serum glucose and urea levels were measured on an enzymatic test using Biosystem S.A. (Barcelona, Spain) reagents.

### 2.3. Statistical Analysis

All numerical data as mean values for each pen were evaluated statistically by one-factor ANOVA using Statistica 12 programme [10]. Differences of mean values between treatments were evaluated by Duncan test. Differences were considered significant at *p* < 0.05 and *p* < 0.01. For analysis the following experimental model was used:y_ij_ = µ + a_i_ + e_ij,_(1)
where y_ij_ is mean value of observed depend variable, µ is mean value of a population, a_i_ influence of treatment, and e_ij_ influence of random factors.

## 3. Results

Chemical composition of wheat and hybrid barley are presented in Table 2. Crude protein content in hybrid barley amounted to 11.02%, while the respective value in wheat was 12.20%. Crude fiber level was estimated at 5.82% in hybrid barley and at 3.11% in wheat.

Amino acid index in hybrid barley amounted to 65 and was higher than in wheat, in which it reached the value of 59. The levels of the most important exogenous amino acids were comparable. Hybrid barley contained by 0.4 g·kg^−1^ more tryptophan.

The nutritional value of the diet is presented in Table 3. Replacement of barley by wheat increased metabolic energy value of a ration by 0.5 MJ·kg^−1^ (calculated on the basis of energy values of components), crude protein by 9 g·kg^−1^, and lysine content by 0.35 g·kg^−1^. On the other hand, high barley content in the diet resulted in an increased fiber level and tryptophan content (by 23 and 0.3 g·kg^−1^, respectively, compared with a diet based entirely on wheat). Amino acid level satisfied the requirements of fattening pigs. The diets were characterized by a balanced, correct lysine-to-metabolic energy ratio. In addition, the contents of exogenous amino acids relative to lysine (assumed to be 100) agreed with nutritional recommendations [8]. In compliance with the experimental design, the energy values of the experimental diets were not supplemented by fats to avoid introducing an additional source of unsaturated fatty acids.

Table 4 shows the production and slaughter yields results of fattening pigs. Initial BW of fattening pigs ranged from 54 to 56 kg (mean 55 kg ± 0.84). After a 78-day fattening period, the pigs from all three groups achieved a similar BW (ca. 120 kg ± 1.4; *p* = 0.085). ADG in all the groups amounted to above 800 g (±12.77) and were not statistically significantly different (*p* > 0.05).

In the present experiment, FI (per one fattening pig) was similar and ranged from 179 to 184 kg (from 2.32 to 2.35 kg·day^−1^ per head). FCR per 1 kg of body weight gain (BWG) was from 2.77 to 2.82 kg and did not differ significantly (*p* > 0.05). The fattening pig fed the diet containing 80% wheat showed slightly lower (ca. 2%) FCR. The lower FCR can be attributed to a higher energy concentration in the diet. The level of nutrition has a significant impact on the animals’ weight gains [11]. A linear relationship is observed between FI and animal growth. Pigs increase their ADFI to get a similar energy intake [12]. Nitikanchana et al. [13] showed that increasing dietary energy density resulted in greater ADG if dietary lysine was at the level of values recommended. Therefore, slightly higher weight gains in the group fed with high wheat content, cannot be a result of higher FI but better utilization of the feed nutrients [14,15].

In the present study, relatively high meatiness at slaughter weight of ca. 120 kg was achieved (ca. 54%; *p* > 0.05). The obtained result was not dependent on the applied diet. Analysis of production performance data showed no significant differences between treatments (but only slightly greater gains (ca. 2%), slightly lower FCR (ca. 2%) and slightly higher meatiness (ca. 0.5%) with increasing wheat percentage content in the diet).

Serum lipid indices are presented in Table 5. Replacing wheat with hybrid barley in the feed dose reduced the serum cholesterol concentration. The level of this component in group I fed the barley-based diet amounted to 2.08 mmol∙dm^−3^, which was significantly lower (*p* < 0.05) compared with groups II and III, in which total cholesterol concentration ranged from 2.37 to 2.49 mmol∙dm^−3^.

Low density lipoprotein (LDL) fraction also significantly varied between the groups I and III (*p* < 0.05). The highest level of this component in serum was noted in group III at the level of 1.18 mmol∙dm^−3^, while in group I it amounted 1.05 mmol∙dm^−3^.

Serum concentration of triglycerides also significantly decrease with increasing hybrid barley content in the diet (*p* < 0.05). In groups I and II it ranged from 0.50 to 0.59 mmol∙dm^−3^, while in group III the level of this component reached 0.72 mmol∙dm^−3^. The obtained results indicate the conspicuous relationships resulting from the use of different basic energy sources, i.e., hybrid barley and wheat in pig feeding.

In the present study, HDL fraction content increased with the higher share of hybrid barley in the feed dose. In group I HDL level reached 1.04 mmol∙dm^−3^ and in group II, 0.97 mmol∙dm^−3^, while in group III their level was significantly lower, amounting 0.84 mmol∙dm^−3^ (*p* < 0.01).

Table 6 presents chosen biochemical indices determined in serum, in particular characterizing nitrogen metabolism, which was similar in all the groups (*p* > 0.05). No significant differences were noted between different basic energy sources.

## 4. Discussion

The nutritional values of hybrid barley and wheat were within the standard range recommended by the Nutrient Requirements for Swine [8]. The crude protein content in hybrid barley was lower than in wheat. On the other hand, wheat always has a higher energy value than barley due to a greater starch and a lower fiber. According to Marin et al. [16], hybrid barley used for pig feeding contained 11.69% of crude protein and crude fiber level was estimated at 5.89%.

The amino acids (AA) content showed that the tryptophan (Trp) concentration was higher compared with wheat grains. Tryptophan is an indispensable AA often limiting pig growth. The dietary Trp is a precursor of serotonin synthesis, which is responsible for feed intake [17]. Therefore, it may act as a regulator of FI by enhancing serotonin signaling in brain. High Trp intake increases feed consumption, which is partly attributed to increased serotonin synthesis [18,19,20].

The prepared diets were characterized by parameters complying with the experimental objectives. The nutritional value of the diets remained in agreement with valid nutritional recommendations [21].

Different cultivars and agronomic conditions cause variation in nutrient composition of wheat and barley grain, that could affect growth performance in pigs. Complete replacement of wheat by barley in pig diets generally reduced their growth and feed intake [22]. However, in the present study, the use of hybrid barley as the basic dietary ingredient for fattening pigs provided similar production parameters as those obtained with diets providing wheat. The results of growth performance were similar in all experimental groups. Daza et al. [23] also did not show significant differences in the final BW, ADG, and FCR between gilts fed a diet containing 98.5% of granulated barley and those fed with 64.37% barley and 10% wheat (control diet); although significant differences were found in ADFI. Gilts (from 100 to 130 kg of BW) fed control diet consumed 2.4% more feed (*p* < 0.001) and tended to grow 11% faster (*p* = 0.008) than gilts fed barley diet [23]. This can be related to possible ability of swine to increase feed intake to compensate the reduced dietary colorific value [22]. Another study [24] showed that the substitution of a conventional feed by a dietary barley reduced ADG and increased ADFI in gilts (from 45 to 92 kg of BW). On the other hand, the study [25] did not reveal differences in ADG and ADFI between gilts (from 86 to 130 kg of BW) fed only granulated barley and those fed a control diet.

In the present experiment, no significant differences were noted between treatments in respect to ADFI and FCR. Daza et al. [25] suggested that FI in pigs fed a diet containing 43% barley and 30% wheat was similar to that for animals fed a barley-based diet. However, FCR was lower by 13% in barley fed groups [25]. The latter study noted an increase in FCR [24].

The use of hybrid barley-based diet in pigs can be designed to obtain the comparable performance results as for animals fed a wheat-based diet. Wheat grains are usually used for the production of a very-high-nutritional value feed mixture. This means that by replacing wheat with hybrid barley it is possible to reach also a high-quality nutritional value of feed mixture. Slight differences in the lower energy level in the diets (12.7 vs. 13.2 MJ ME) did not affect FCR. Although, it is known that the lower FCR can be attributed to a higher energy concentration in the diet, which can be found for example for corn-based feeds [26]. Lower energy value of barley than of that in wheat grain was due to its greater dietary fiber content, that could reduce also swine growth performance [26].

Analysis of data of the production performance showed no differences in weight gain and meatiness with increasing content of wheat in the diet. Similar results were reported by [23,24,25], who replaced dietary wheat with granulated barley. Daza et al. [23] observed that the treatments had no effect on carcass traits except sirloin weight, which was lower in gilts fed barley diet than in those fed control diet. Similarly, Daza et al. [24] noted a lower loin weight in pigs fed barley than those fed a conventional feed, but no differences were found for ham and foreleg weights and percentage of the cuts in carcass. Additionally, Daza et al. [25] found similar results in heavy pigs fed mainly barley during growing-finishing period. Senčić et al. [27] reported that meatiness importantly was associated with BW (pigs slaughtered at lower BW had significantly higher meatiness). In our study, pigs slaughtered had similar BW that is why the meat content was also comparable. Slightly higher lean meat content and growth rate and better feed conversion by fattening pigs fed a wheat-based diet could result from somewhat higher energy availability in this group and better digestibility. On the other hand, Turyk et al. [28] used concentrated feed with 80% triticale grain and reported leanness higher by 1% (*p* < 0.05) compared with a barley grain-based diet. In a feeding trial by Banaszkiewicz et al. [29], in which triticale was used instead of barley in the diet for growing and finishing pigs, a tendency to increase the meat content in swine fed barley was shown (57.2% and 60.1%).

Using the feed mixtures of high-quality and high nutritional value, irrespective of the grain source, allowed to obtain a good meatiness without any difference between groups. In spite of a slightly lower level of crude protein in the diet with 80% of hybrid barley compared to 80% of wheat grain (155 vs. 164.5 g kg^−1^) the collected results were similar. It could indicate that keeping Lys content/ME ratio, which was on the same level in the present study, allowed to maintain similar pig growth performance. A comparable relationship was observed by other authors [22].

Total blood cholesterol content depends mostly on genetic traits (breed, sex), diet, and endogenous synthesis of this compound in the liver [30,31]. Total serum cholesterol levels determined in the present experiment were slightly raised compared to reference range (according to Winnicka [32] total cholesterol reference limits in pigs range from 0.5 to 2.1 mmol∙dm^−3^).

Application of barley in swine diets usually increases dietary fiber content. Barley contains β-glucans (included in a soluble dietary fiber), which can be easily fermented by the gut microflora and stimulated the production of butyrate, that is the main source of energy for colonocytes, what moreover affects the health status of animals [22]. Dietary fiber, especially β-glucans, in barley decreases level of cholesterol and LDL-cholesterol in human [6] and animals [33,34]. Significant barley content in pig diet also reduces cholesterol level even by up to 18% since it contains inhibitors of LDL synthesis in liver [35]. Other animal studies [36,37] also demonstrated that barley reduces total cholesterol level. This cholesterol reduction phenomenon was explained by binding and excretion of bile acids, hampered lipid absorption, the presence of short-chain fatty acids or changes in insulin concentration [38,39,40,41]. The most popular theory claims that barley increases fecal excretion of bile acids synthesized in the body from cholesterol. Binding and excretion of bile acids results in directing increasingly larger amounts of cholesterol accumulated in the body to their synthesis, thereby reducing its concentration in serum [38]. It also appears irrefutable that changes in viscosity of the intestinal content play a role in this process, and viscosity rise can lead to the reduction of cholesterol absorption and to binding of fatty acids [37]. The increased intestinal content viscosity limits bile acid return to the liver or even leads to their complete absorption [39,40,41].

Studies in humans and animals have revealed that barley specifically reduces total cholesterol and LDL lipoprotein content [36,42] with a concomitant increase in HDL cholesterol fraction. It could be related to the presence of beta-glucan in barley. A similar effect was noted in results of Nicolosi et al. [43] with application of β-glucan from yeasts. The yeast-derived β-glucan fiber significantly lowered total cholesterol concentrations and at the same time HDL-cholesterol concentrations rose in men. In the present research, the content of HDL fractions also increased significantly. Other relationships were observed in the study [28] designed to check whether feeding of barley or triticale to fattening pigs (from 30.5 to about 110 kg of BW) influences serum lipid indices. The pigs fed with more than 80% of barley contained in diets showed statistically nonsignificant increases in the level of total cholesterol and triglycerides but also in HDL and LDL fractions. In the present study, the levels of total cholesterol, LDL fraction, and triglycerides decreased with increasing hybrid barley content in the ration while HLD-cholesterol level rose. The obtained results may indicate that lipid metabolism in the body depends on energy source. Reducing the level of energy and protein in the feed decreased the concentration of total cholesterol and LDL fraction [14]. A similar relationship was noted by Sawosz et al. [11], showing only a tendency towards decreasing the level of triglycerides, total cholesterol, and LDL fraction in pigs fed low energy and protein diet. Unfortunately, also the concentration of HDL decreased in that group. Da Costa et al. [44] found that diets with reduced energy and protein contents increased potentials for substrate (protein, glycogen, and lipid) turnover and mitochondrial function in growing pigs. Więcek et al. [14] did not find an unambiguous, statistically confirmed impact of the increased energy in the feed (by introducing the addition of 4% linseed oil to the diet) on the level of total cholesterol and its fractions. In the event of an excess of energetic substrates in the body, then fatty acids are esterified and enter the blood in the form of very low-density lipoproteins [45]. A similar relationship as in the present study was observed by other authors [46,47,48] when barley was replaced by corn (LDL level was lower by 18% in animals receiving dietary barley). Additionally, Kalra and Jood [7], who used flour of three barley cultivars in the diets of rats, noted significant reductions in the level of total cholesterol, LDL-cholesterol, and triglyceride, and significant increase in the level of HDL-cholesterol in serum compared with the casein diet.

The collected results showed that the high level of hybrid barley in diet had beneficial effect on the lipid indicators in blood serum. Usage diet containing 80% of hybrid barley indicated a good animal health status. It means that the level of cholesterol and its fractions can be modified by the diet and also correlated to the amount of dietary barley.

All estimated biochemical indices determined in serum were within normal range [32], which can indicate good health status of animals. In human medicine, well-documented studies have revealed that one of the benefits of a barley-rich diet is its ability to flatten the postprandial glucose and insulin curves [49,50]. A high feed consumption increases the insulin concentration in the serum of pigs [51]. Insulin elevates utilization of glucose for triglycerides synthesis during higher fat deposition in the animal’s body [11]. When the blood glucose level decreases, then the fatty acids, as a reserve material, are used by both the muscles and the liver [45]. Our study did not confirm reduction of serum glucose level. Additionally, β-glucan supplementation resulted in a lower number of pathogens found in liver what can be related with reduced intestinal translocation of bacteria [52].

Animals fed with the reduced energy and protein level in feed showed a tendency towards lower total protein, albumin, and globulin content [14]. Our study did not confirm that result. The concentration of total protein and albumin were within the same range for all the experimental groups.

Dietary β-glucans may also reduce protein fermentation [22]. Literature data indicate that the slightly greater fiber and lower protein content in diets with barley may increase gut health of pigs by reducing intestinal pH and ammonia production [53]. In the present study, no differences of urea in blood serum were noted probably because the level of crude protein in all diets was similar. Reduction of protein content in the feed mixture (with appropriate amino acid balance) reduces the urea concentration in blood serum and decreases nitrogen excretion in urine, without deterioration of fattening results and carcass quality [54]. A study [55] showed that by lowering the protein level in the diet by 15%, a 35% reduction in serum concentration of urea can be achieved in fattening pigs. Another study [56] also reported reductions in urea values (5–8%). Urea is a compound formed by the decomposition of proteins in the body, and a too high level in the blood serum indicates that animals do not use feed amino acids, which may be caused by too high protein content in the feed dose [57].

## 5. Conclusions

The use of 80% hybrid barley as the basic ingredient of diets for fattening pigs (from 55 to 120 kg body weight) provided similar production parameters as those obtained with 80% wheat. Data collected in all the experimental groups were similar (daily gain of fattening pigs of 800 g, feed conversion 2.81 kg·kg^−1^, and meatiness of ca. 54%). All the animals were characterized by good health status. In addition, the diet with high level of barley had a beneficial effect on blood lipid indices. Therefore, hybrid barley can be recommended for use as the only cereal ingredient in diets for fattening pigs.

## Figures and Tables

**Table 1 animals-10-01987-t001:** Design of experiment.

Item	Treatment		
	Hybrid Barley 80%	Hybrid Barley 40% Wheat 40%	Wheat—80%
Number of animals in group	48	48	48
Number of pens in group	6	6	6

**Table 2 animals-10-01987-t002:** Chemical composition and amino acid index wheat and hybrid barley.

Item	Components	
	Hybrid Barley	Wheat
**Nutrients, %·kg^−1^**		
Dry matter	91.14	91.1
Crude ash	2.1	1.9
Crude protein	11.02	12.2
Crude fiber	5.82	3.11
Crude fat	1.23	2.32
**Amino acids, g·kg^−1^**		
Lysine	3.68	3.88
Methionine + Cystine	4.11	4.6
Tryptophan	1.42	1.01
Arginine	4.68	5.18
Histidine	2.49	2.52
Phenylalanine	5.55	5.57
Tyrosine	2.11	3.02
Leucine	7.49	7.6
Isoleucine	3.56	3.94
Valine	4.99	4.92
Alanine	4.17	3.48
Glycine	4.32	4.21
Proline	12.06	11.87
Threonine	3.64	3.63
Serine	4.6	4.51
Glutamic acid	25.15	20.11
**Amino acid index**	65	59

**Table 3 animals-10-01987-t003:** Chemical composition and nutritive value of the diets for fattening pigs.

Item	Treatment		
	Hybrid Barley—80%	Hybrid Barley—40% Wheat—40%	Wheat—80%
**(in %)**			
Hybrid barley	84.8	42.2	-
Wheat	-	42.2	84.05
Soybean meal	10.6	11	11.35
Premix *	2.5	2.5	2.5
Fodder yeast	2	2	2
Optazyme Lidermix^®^ **	0.1	0.1	0.1
**Nutritive value (g in 1 kg of mixture)**			
Metabolizable Energy, MJ	12.7	12.9	13.2
Crude protein	155	158.1	164.5
Crude fat	11.7	16.3	20.9
Crude fiber	53.4	41.9	30.5
Lysine	9.14	9.32	9.49
Methionine	2.84	2.9	2.96
Threonine	5.45	5.51	5.56
Tryptophan	1.96	1.81	1.66
Ca	4.43	4.57	4.7
P	5.36	5.37	5.37
Na	1.64	1.64	1.64
**Content crude protein (g) and lysine (g) per 1 MJ ME**			
Crude protein/Metabolizable energy	12.2	12.2	12.5
Lysine/Metabolizable energy	0.73	0.72	0.73
**Amino acids in ratio to lysine content**			
Methionine + Cystine	64	64	65
Threonine	60	59	59
Tryptophan	21	19	18

* Premix (2.5%)—content in 1 kg: vitamin A—400,000 IU; vitamin D3—90 IU; vitamin E—3500 mg; choline—3000 mg; betaine—20,000 mg; Ca—147.357 g; P—55.000 g; Na—60.000 g; Cl—93.323 g; lysine—94 g; Methionine—20 g. ** OPTAZYME LIDERMIX®—premix contain enzymes for swine, content in 1 kg: endo-1,4 β-xylanase 1 100,000 units; endo-1,3(4) β-glucanase 100,000 units AGL.

**Table 4 animals-10-01987-t004:** Performance and slaughtering yield results of fattening pigs.

Specification	Treatment				
	Hybrid Barley—80%	Hybrid Barley—40% Wheat—40%	Wheat—80%	SEM	*p*-Value
**Performance results**					
Body weight (BW), kg					
initial	54.39	56.04	55.52	1.607	0.358
final	119.48	119.66	121.97	1.711	0.085
Average daily body weight gains (ADG), g·day^−1^	834	826	851	5.894	0.174
Average daily feed intake (ADFI) per head, kg·day^−1^	2.34	2.32	2.35	2.145	0.774
Feed conversion ratio (FCR), kg feed∙kg BWG^−1^	2.81	2.82	2.77	0.026	0.349
**Average results of slaughter yield of fatteners**					
Cold carcass weight, kg	88.79	88.48	89.36	1.366	0.727
Fat thickness, mm	20.13	21.97	17.90	0.899	0.403
Muscle thickness, mm	55.34	56.13	59.93	1.146	0.110
Slaughter yield, %	74.30	73.74	73.25	1.123	0.973
Meatiness, %	54.00	54.35	54.52	0.809	0.683

**Table 5 animals-10-01987-t005:** Lipid indicators determined in blood serum of fatteners.

Specification	Treatment			SEM	*p*-Value
	Hybrid Barley—80%	Hybrid Barley—40% Wheat—40%	Wheat—80%		
Total cholesterol, mmol∙dm^−3^	2.08 ^a^	2.37 ^b^	2.49 ^b^	1.105	0.021
HDL, mmol∙dm^−3^	1.04 ^A^	0.97 ^A^	0.84 ^B^	0.035	0.001
LDL, mmol∙dm^−3^	1.05 ^a^	1.10 ^ab^	1.18 ^b^	0.033	0.026
Triglycerides, mmol∙dm^−3^	0.50 ^a^	0.59 ^a^	0.72 ^b^	0.056	0.013

Differences in rows signed with different subscript letters significant by: A,B—*p* ≤ 0.01 and a,b—*p* ≤ 0.05.

**Table 6 animals-10-01987-t006:** Biochemical indicators determined in blood serum of fatteners.

Specification	Treatment			SEM	*p*-Value
	Hybrid Barley—80%	Hybrid Barley—40% Wheat—40%	Wheat—80%		
Total protein, g∙dm^−3^	63.64	62.43	64.96	1.08	0.752
Albumin, g∙dm^−3^	31.16	31.26	32.54	0.9	0.508
α-globulin, g∙dm^−3^	13	12.6	13.42	0.4	0.368
β-globulin, g∙dm^−3^	11.9	11.9	12.3	0.41	0.714
γ-globulin, g∙dm^−3^	7.58	6.67	6.7	0.58	0.516
Glucose, mmol∙dm^−3^	4.5	4.3	4.35	0.29	0.864
Urea, mmol∙dm^−3^	4.1	4	4.05	0.19	0.936

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
