# Peer review of "Effect of Different Amounts of Hybrid Barley in Diets on the Growth Performance and Selected Biochemical Parameters of Blood Serum Characterizing Health Status in Fattening Pigs"

_animals, 2020, doi:10.3390/ani10111987_

Round 1

Reviewer 1 Report

General comments:

The authors are to be congratulated on producing a valuable paper. The study was well conducted, and the results would be of interest to the scientists involved in feed ingredients evaluation and for those involved in the practical application of canola meal in animal and swine nutrition.

Overall, I think the Discussion section of the manuscript needs to be more improved. I suggest the authors would refer to the valid results of other studies to discuss the results of the present study. Also, the authors simply listed the results of the previous studies and compared them to the results of the present study. The authors need to discuss possible or reasonable reasons for the observations of the current study by comparing the results of the previous studies. It is hard to discuss the results with a novel ingredient, but authors need to discuss the observations and present scientific reasons.

Line 202-217 : Growth performance

Line 252-279 : Blood cholesterols and triglycerides.

Line 288-289 : The reference is not that valid to discuss the result of the present study.

Line 290-294 : Parameters in Table 6.

Line 295 - 302: Blood urea.

Statistical analysis is conducted in order to assist the decision if differences are due to natural variation or a treatment effect, and if the P-value is higher than a predefined threshold, it is interpreted as no effect. If pertinent, trends (0.05 ≤ P < 0.10) are also reported. In the manuscript, authors used “tendency or tend” several times with no significant difference (P>0.10) and numerical difference. It can be a rather unscientific and subjective explanation of the results.

Line 25

Line 218 - 219

Line 290, 292

Specific comments:

Line 67-68: The distribution of sex of pigs in the pens should be mentioned

Line 95; “average daily gain (ADG)”

Line 192 : Please change “this AA” to “Trp”

Line 206 : Please delete “tended to”.

Line 307 : Please rewrite the statement in terms of blood parameters.

Table 5 : please synchronize the unit (abstract: mmol l-1; table 5,6:mmol∙dm-3) of the blood parameters and use the correct unit as per the guideline of the journal.

References : Please check the format of all references as per the guideline of the journal.

For examples,

 Line327-328: No doi. (https://doi.org/10.1093/ajcn/80.5.1185).

 Line336-338: No doi. (https://doi.org/10.22358/jafs/70588/2005)

Author Response

Thank you for your reviews.

The data has been completed.

The discussion has been improved and supplemented with elements that were suggested by the Reviewer.

General comments:

The authors are to be congratulated on producing a valuable paper. The study was well conducted, and the results would be of interest to the scientists involved in feed ingredients evaluation and for those involved in the practical application of canola meal in animal and swine nutrition.

Overall, I think the Discussion section of the manuscript needs to be more improved. I suggest the authors would refer to the valid results of other studies to discuss the results of the present study. Also, the authors simply listed the results of the previous studies and compared them to the results of the present study. The authors need to discuss possible or reasonable reasons for the observations of the current study by comparing the results of the previous studies. It is hard to discuss the results with a novel ingredient, but authors need to discuss the observations and present scientific reasons.

Line 202-217 : Growth performance

Line 221-229

The use of hybrid barley-based diet in pigs can be designed to obtain the comparable performance results as for animals fed wheat-based diet. Wheat grains are usually used for the production of a very-high-nutritional value feed mixture. This means that replacing wheat by hybrid barley is possible to reach also a high-quality nutritional value of feed mixture. Slight differences in the lower energy level in the diets (12.7 MJ EM vs 13.2 MJ EM) did not affect FCR. Although it is known that the lower FCR can be attributed to a higher energy concentration in the diet, which can be found for example for corn-based feeds [26]. Lower energy value of barley than of that in wheat grain was due to its greater dietary fiber content, that could reduce also swine growth performance [26].

Line 247-253

Using the feed mixtures of high-quality and high nutritional value, irrespective of the grain source, allowed to obtain a good meatiness without any difference between groups. In spite of a slightly lower level of crude protein in the diet with 80% of hybrid barley compared to 80% of wheat grain (155 g kg-1 vs 164,5 g kg-1) the collected results were similar. It could indicate that keeping Liz content / EM ratio, which was on the same level in the present study, allowed to maintain similar pig’s growth performance. A comparable relationship was observed by other authors [22].

Line 252-279 : Blood cholesterols and triglycerides.

Line 303-306

The collected results showed that the high level of hybrid barley in diet had beneficial effect on the lipid indicators in blood serum. Usage diet containing 80% of hybrid barley indicated a good animal health status. It means that the level of cholesterol and its fractions can be modified by the diet and also correlated to the amount of dietary barley.

Line 288-289 : The reference is not that valid to discuss the result of the present study.

Done.

Line 290-294 : Parameters in Table 6.

Line 295 - 302: Blood urea.

Line 317-333

Animals fed with the reduced energy and protein level in feed showed a tendency towards lower total protein, albumin and globulin content [14]. Our study did not confirm that result. The concentration of total protein and albumin were within the same range for all the experimental groups. On the other hands, the addition of 4% linseed oil to the diet for fatteners increased the level of total protein and albumin [14].

Dietary β-glucans may also reduce protein fermentation [22]. Literature data indicate that the slightly greater fiber and lower protein content in diets with barley may increase gut health of pigs by reducing intestinal pH and ammonia production [53]. In the present study, no differences of urea in blood serum were noted probably because the level of crude protein in all diets was similar. Reduction of protein content in the feed mixture (with appropriate amino acid balance) reduces the urea concentration in blood serum and decreases nitrogen excretion in urine, without deterioration of fattening results and carcass quality [54]. Study [55] showed that by lowering the protein level in the diet by 15%, a 35% reduction in serum concentration of urea can be achieved in fattening pigs. Other study [56] also reported reductions in urea values (5 to 8%). Urea is a compound formed by the decomposition of proteins in the body, and it’s too high level in the blood serum indicates that animals do not use feed amino acids, which may be caused by too high protein content in the feed dose [57].

Statistical analysis is conducted in order to assist the decision if differences are due to natural variation or a treatment effect, and if the P-value is higher than a predefined threshold, it is interpreted as no effect. If pertinent, trends (0.05 ≤ P < 0.10) are also reported. In the manuscript, authors used “tendency or tend” several times with no significant difference (P>0.10) and numerical difference. It can be a rather unscientific and subjective explanation of the results.

Line 25

Line 218 - 219

Line 290, 292

Done.

Specific comments:

Line 67-68: The distribution of sex of pigs in the pens should be mentioned

Done (trying to keep sex ratio 1:1)

Line 95; “average daily gain (ADG)”

Done.

Line 192 : Please change “this AA” to “Trp”

Done.

Line 206 : Please delete “tended to”.

Done.

Line 307 : Please rewrite the statement in terms of blood parameters.

Table 5 : please synchronize the unit (abstract: mmol l-1; table 5,6:mmol∙dm-3) of the blood parameters and use the correct unit as per the guideline of the journal.

Done.

References : Please check the format of all references as per the guideline of the journal.

For examples,

 Line327-328: No doi. (https://doi.org/10.1093/ajcn/80.5.1185).

 Line336-338: No doi. (https://doi.org/10.22358/jafs/70588/2005)

Done.

Reviewer 2 Report

The authors did a good work improving the manuscript in agreement with the comments received in the reviews of the first submission. I can recommend acceptance for publication. Thanks for the work done.

Author Response

Thank you for your reviews.

Best regards

Anna Szuba-Trznadel

Round 2

Reviewer 1 Report

Line 109: Please add the materials to collect the blood. As per the title of Table 6 (Biochemical indicators determined in blood and blood serum of fatteners), you analyzed the samples for blood and serum samples, respectively. Please present the information on the collection tubes. And if you centrifuged the blood samples to get the serum, please add the details with the centrifuge condition and machine.

Line 250: change "Liz content /ME ratio" to "Lys content / ME ratio"

Line 283: Revise "but also in high-(HDL) and low-density lipoprotein (LDL) fractions." to "but also in HDL and LDL fractions."

Line 285: change " HLD level rose" to "HDL-cholesterol level rose".

Line 319-320 : Delete the sentence "On the other hands, the addition of 4% linseed oil to the diet for fasteners increased the level of total protein and albumin [14]. This statement with reference is not correct to the discussion for blood protein, ablumin, and globulin concentrations.

Line 334-335: please indicate the inclusion level of barley and wheat. For instance, "the use of 80% hybrid barley" "those obtained with 80% wheat".

Line 334-340: Please rewrite the conclusion. The last sentence (line 339-340; In addition, the diet with high level of barley, had a beneficial effect on blood lipid indices) should be presented before the final conclusion (Line 338-339; Therefore, hybrid barley can be recommended for use as the only cereal ingredient in diets for fattening pigs).

Table 4, 6: There is no subscripts in the tables. Please delete the footnotes in table 4 and 6.

Table 4: Revise "Yield" with "Slaughter yield" or "Carcass yield".

Table 5: In the footnote, you mentioned that differences in rows signed with different subscript letters significant by: A, B – p ≤ 0.01 and a,b – p ≤
171 0.05. I recommended you to use superscript instead of the subscript. 

Please recheck the conflicts of interest to make sure. Because the hybrid barley (HyvidoTM) used in the present study was provided by the company (Syngenta Co., Warszawa, Poland). 

Author Response

Response to Reviewer 1 Comments

Thank you for your reviews.

Line 109: Please add the materials to collect the blood. As per the title of Table 6 (Biochemical indicators determined in blood and blood serum of fatteners), you analyzed the samples for blood and serum samples, respectively. Please present the information on the collection tubes. And if you centrifuged the blood samples to get the serum, please add the details with the centrifuge condition and machine.

The blood was taken into Starstedt type-tubes, which contains bead coated with a clotting activator and then samples were centrifuged using MPW-223e laboratory centrifuge.

Biochemical indicators and lipid indicators were determined in blood serum of fatteners.

Line 250: change "Liz content /ME ratio" to "Lys content / ME ratio"

done

Line 283: Revise "but also in high-(HDL) and low-density lipoprotein (LDL) fractions." to "but also in HDL and LDL fractions."

done

Line 285: change " HLD level rose" to "HDL-cholesterol level rose".

done

Line 319-320 : Delete the sentence "On the other hands, the addition of 4% linseed oil to the diet for fasteners increased the level of total protein and albumin [14]. This statement with reference is not correct to the discussion for blood protein, ablumin, and globulin concentrations.

done

Line 334-335: please indicate the inclusion level of barley and wheat. For instance, "the use of 80% hybrid barley" "those obtained with 80% wheat".

done

Line 334-340: Please rewrite the conclusion. The last sentence (line 339-340; In addition, the diet with high level of barley, had a beneficial effect on blood lipid indices) should be presented before the final conclusion (Line 338-339; Therefore, hybrid barley can be recommended for use as the only cereal ingredient in diets for fattening pigs).

done

Table 4, 6: There is no subscripts in the tables. Please delete the footnotes in table 4 and 6.

done

Table 4: Revise "Yield" with "Slaughter yield" or "Carcass yield".

done

Table 5: In the footnote, you mentioned that differences in rows signed with different subscript letters significant by: A, B – p ≤ 0.01 and a,b – p ≤
171 0.05. I recommended you to use superscript instead of the subscript. 

done

Please recheck the conflicts of interest to make sure. Because the hybrid barley (HyvidoTM) used in the present study was provided by the company (Syngenta Co., Warszawa, Poland). 

Name of the variety of winter hybrid barley is HyvidoTM, which is in the Syngenta Company's sale offer.

None of authors has any financial and personal relationships with other people or  organisations that could inappropriately influence or bias the content of the paper.

Kind regards

Anna Szuba-Trznadel

This manuscript is a resubmission of an earlier submission. The following is a list of the peer review reports and author responses from that submission.

Round 1

Reviewer 1 Report

I manuscript was noticed small inaccuracies

In discussion line 193-195  authors inform that “ Total serum cholesterol levels determined in the present experiment were in the upper normal range (according to Winnicka [24] total cholesterol reference limits in pigs range from 0.5-2.1 mmol ∙1-1)”  

This information does not agree with the data in table 5 where it can be seen that the total cholesterol level in groups II (wheat and hybrid barley were used) and III (contained 80% of wheat) is 2.37 and 2.49 respectively.

This is an inaccurate definition, especially that a statistical significance was observed between groups II and III and group I, which is characterized by values in the reference limits.

The above comments do not diminish the value of work

Author Response

Response to Review 1 Comments

Thank you for your reviews.

The data has been completed.

The discussion has been improved and supplemented with elements that were suggested by the Reviewer.

Comments and Suggestions for Authors

I manuscript was noticed small inaccuracies

In discussion line 193-195  authors inform that “ Total serum cholesterol levels determined in the present experiment were in the upper normal range (according to Winnicka [24] total cholesterol reference limits in pigs range from 0.5-2.1 mmol ∙1-1)”  

Response 1: Line 240-242: Total serum cholesterol levels determined in the present experiment were slightly raised compared to reference range (according to Winnicka [32] total cholesterol reference limits in pigs range from 0.5 – 2.1 mmol∙dm-3).

This information does not agree with the data in table 5 where it can be seen that the total cholesterol level in groups II (wheat and hybrid barley were used) and III (contained 80% of wheat) is 2.37 and 2.49 respectively.

This is an inaccurate definition, especially that a statistical significance was observed between groups II and III and group I, which is characterized by values in the reference limits.

Response 2: Line 164-167: Serum lipid indices are presented in Table 5. Replacing wheat with hybrid barley in the feed dose reduced the serum cholesterol concentration. The level of this component in group I fed the barley-based diet amounted to 2.08 mmol∙dm-3, which was significantly lower (p < 0.05) compared with groups II and III, in which total cholesterol concentration ranged from 2.37 to 2.49 mmol∙dm-3.

Kind regards,

Anna Szuba-Trznadel

Reviewer 2 Report

The aim and experimental design of the present study are reasonable, and the results of the present study can provide information for the use of hybrid-barley to fattening pigs in the swine industry. In order to provide scientific information, this manuscript needs some additional revisions.

  1. Please provide general information on hybrid barley in the part of the introduction, and provide specific information on hybrid barley which is used in the present study in the M&M.

  1. More details of the statistical analysis are needed, in particular statistically significant difference or P-value.

  1. Line 124 “The fattening pig fed the diet containing 80% wheat showed a slightly better feed conversion ratio”. Line 129-130 “analysis of production performance data showed only a tendency towards slightly greater gains, better feed conversion, and higher meatiness with increasing wheat percentage content in the diet. How can you know that? Unfortunately, the p-value for FCR is 0.349 and the p-value for meatiness is 0.683. Statistical analysis is conducted in order to assist the decision if differences are due to natural variation or a treatment effect, and if the P-value is higher than a predefined threshold, it is interpreted as no effect. Therefore, I suggest to never discuss numerical differences, unless they are so big that they would be of practical relevance if they were significant.

  1. The results for growth performance and blood parameters of fattening pigs are insufficient to be published. In the second paragraph, the authors mentioned that barley has a beneficial effect on the quality of pork. In Table 4, authors suggested the result for meatiness. Therefore, authors need to add the data of carcass analysis to supplement the quality of the manuscript.

  1. Please check the information on the references again.

Ex. The name of authors and journal were incorrect on reference 41.

Line 62: Please describe the details of the semi ad libitum feed system

Line 76: In the results (Line 106), the authors mentioned that “calculated on the basis of energy values of components”. It is necessary to clearly indicate the calculated values and analyzed values of the nutritional values of the diets.

Line88: More details of the widely accepted standard methods for total protein and albumins. Also, please explain the abbreviation “BCA”.

Line 89. Reference information for Biosystems reagents is need

Line 102: You used the word “amino acid index” to explain the biological value of protein in the Table 2. Please synchronize the words in the manuscript.

Line 107-108: Please revise the units of the values. Ex. Total protein by 9 g/kg and lysine content 0.35 g/kg. fiber level and tryptophan content (by 23 g/kg and 0.3 g/kg, respectively

Table 4. Please check the unit of the average daily feed intake or ADFI values of the treatments. The ADFI values ranged from 179 to 184 kg per head is not make sense.  

Line 138:
high-density lipoprotein fraction (HDL) content also increased. This sentence is not enough to explain the result completely. Please indicate the subject or object about the increase of HDL-cholesterol.

Line 161: You referred the result of Szuba-Trznadel and Fuchs (2015) that “the concentration of this amino acid in the diet stimulates serotonin synthesis which is responsible for feed intake.

What is the reason that you referred to this reference and discussed the tryptophan content in the hybrid-barley? Did the high content of tryptophan in the hybrid-barley improve the feed intake of pigs? As you know, the contents of lysine, methionine, threonine were lower in the hybrid-barley than in wheat (Table 2). Therefore, I suggest to add an additional discussion about the relationship between tryptophan content of the hybrid barley and feed intake data when you mentioned the effect of tryptophan on the feed intake of pigs

Line 170: Please revise the word “animals”. Please note the specific animal type or stage (ex. pigs or gits).

Line 171. More details of the referred information “although significant differences were found in average daily feed intake”. In the statement, it is hard to understand which diet was greater in ADFI.

Line 172-173. There is no significant difference in the result of FCR. Thus, “80% wheat showed a slightly better feed conversion ratio” is an incorrect statement.

Line 181: “by authors [14, 18], who replaced wheat with granulated barley in the diets.”. More details of the references [14,18] are needed to support your statement.

Line 211: more details of the reference of Nicolosi et al. [35] are needed to support your statement.

Ex. beta-glucan content, which animals, etc.

Line 214: please revise “studies [20]” to “study [20]”

Line 215-217: more details of the reference [20] are needed to support the discussion. Ex. how much triticale fed to the pigs in the study of reference [20]. Also, you need to discuss the result of the present study considering the result of reference [20].

Line 219-220: “The obtained results may indicate that lipid metabolism in the body depends on the energy source.” As per the nutritional values of the diets (Table 3), the energy value and crude fat content in the wheat 80% diet were higher than other diets. These differences may affect serum cholesterol components. I suggest you to openly discuss this difference of the diets.

Line 230: Blood components related to nitrogen metabolism (protein, albumin, globulin, urea) need to be discussed.

Author Response

Response to Review 2 Comments

Thank you for your reviews.

The data has been completed.

The discussion has been improved and supplemented with elements that were suggested by the Reviewer.

Comments and Suggestions for Authors

The aim and experimental design of the present study are reasonable, and the results of the present study can provide information for the use of hybrid-barley to fattening pigs in the swine industry. In order to provide scientific information, this manuscript needs some additional revisions.

Please provide general information on hybrid barley in the part of the introduction, and provide specific information on hybrid barley which is used in the present study in the M&M.

Response 1: General information on hybrid barley was provide:

Line 39-43: Field observations show that under good weather conditions, the yield of hybrid barley is even 14% higher compared to conventional varieties. Moreover, hybrid varieties are characterized by a higher content of total protein. They also have only slightly lower energy value than wheat because hybrid barley contains less crude fiber than conventional varieties (www.syngenta.pl).

Line 72: Hybrid varieties of barley HyvidoTM (Syngenta Co., Warszawa, Poland) was used in the study.

More details of the statistical analysis are needed, in particular statistically significant difference or P-value.

Response 2: Line 122: Differences were considered significant at p < 0.05 and p < 0.01.

Line 124 “The fattening pig fed the diet containing 80% wheat showed a slightly better feed conversion ratio”. Line 129-130 “analysis of production performance data showed only a tendency towards slightly greater gains, better feed conversion, and higher meatiness with increasing wheat percentage content in the diet. How can you know that? Unfortunately, the p-value for FCR is 0.349 and the p-value for meatiness is 0.683. Statistical analysis is conducted in order to assist the decision if differences are due to natural variation or a treatment effect, and if the P-value is higher than a predefined threshold, it is interpreted as no effect. Therefore, I suggest to never discuss numerical differences, unless they are so big that they would be of practical relevance if they were significant.

Response 3: Done

Line 150-152: FCR per 1 kg of body weight gain (BWG) was from 2.77 kg to 2.82 kg and did not differ significantly (p > 0.05). The fattening pig fed the diet containing 80% wheat showed slightly lower (ca. 2%) FCR.

Line 160163: Analysis of production performance data showed no significant differences between treatments (but only a tendency towards slightly greater gains (ca. 2%), slightly lower FCR (ca. 2%) and slightly higher meatiness (ca. 0.5%) with increasing wheat percentage content in the diet).

The results for growth performance and blood parameters of fattening pigs are insufficient to be published. In the second paragraph, the authors mentioned that barley has a beneficial effect on the quality of pork. In Table 4, authors suggested the result for meatiness. Therefore, authors need to add the data of carcass analysis to supplement the quality of the manuscript.

Response 4: The data of slaughtering yields results of fattening pigs were added in Table 4.

Please check the information on the references again.

Ex. The name of authors and journal were incorrect on reference 41.

Response 5: Done

Line 62: Please describe the details of the semi ad libitum feed system

Response 6: Line 70-71: A semi ad libitum feeding system was used (the amount of the mixture given to pigs was increased depending on the individual intake of the daily feed dose).

Line 76: In the results (Line 106), the authors mentioned that “calculated on the basis of energy values of components”. It is necessary to clearly indicate the calculated values and analyzed values of the nutritional values of the diets.

Response 7: Nutrients (according to chemical analyzed) were presented in Table 2. The energy value (calculated to Nutrient Requirements for Swine) was moved and described in the Material and methods.

Line88: More details of the widely accepted standard methods for total protein and albumins. Also, please explain the abbreviation “BCA”.

Response 8: Line 115-116: The BCA (Bicinchoninic Acid) Protein Assay Kit (Sigma-Aldrich, USA) was used for quantitation of total protein (TP)

Line 89. Reference information for Biosystems reagents is need

Response 9: Line 117-118: Serum glucose and urea levels were measured on an enzymatic test using Biosystem S.A. (Barcelona, Spain) reagents.

Line 102: You used the word “amino acid index” to explain the biological value of protein in the Table 2. Please synchronize the words in the manuscript.

Response 10: Done

Line 107-108: Please revise the units of the values. Ex. Total protein by 9 g/kg and lysine content 0.35 g/kg. fiber level and tryptophan content (by 23 g/kg and 0.3 g/kg, respectively

Response 11: Done

Table 4. Please check the unit of the average daily feed intake or ADFI values of the treatments. The ADFI values ranged from 179 to 184 kg per head is not make sense.  

Response 12: Done

Average daily feed intake (ADFI) per head, kg·day-1 was showed in Table 4.

Line 138:
high-density lipoprotein fraction (HDL) content also increased. This sentence is not enough to explain the result completely. Please indicate the subject or object about the increase of HDL-cholesterol.

Response 13: Done

Line 179-180: HDL fraction content increased with the higher share of hybrid barley in the feed dose.

Line 161: You referred the result of Szuba-Trznadel and Fuchs (2015) that “the concentration of this amino acid in the diet stimulates serotonin synthesis which is responsible for feed intake.

What is the reason that you referred to this reference and discussed the tryptophan content in the hybrid-barley? Did the high content of tryptophan in the hybrid-barley improve the feed intake of pigs? As you know, the contents of lysine, methionine, threonine were lower in the hybrid-barley than in wheat (Table 2). Therefore, I suggest to add an additional discussion about the relationship between tryptophan content of the hybrid barley and feed intake data when you mentioned the effect of tryptophan on the feed intake of pigs

Response 14: Done

Line 196-199: Trp is an indispensable AA that is often limiting growth in pigs. Trp may act as a regulator of FI by enhancing serotonin signaling in the brain, because it is a precursor for serotonin synthesis. High Trp intake increases feed consumption, which is partly attributed to increased serotonin synthesis.

Line 170: Please revise the word “animals”. Please note the specific animal type or stage (ex. pigs or gits).

Response 15: Done

Line 171. More details of the referred information “although significant differences were found in average daily feed intake”. In the statement, it is hard to understand which diet was greater in ADFI.

Response 16: Line 205-213: The study of Daza et al. [22] also did not show statistically significant differences in the final BW, ADG and FCR between gilts fed a diet containing 98.5% of granulated barley and those fed with 64.37% barley and 10% wheat (control diet); although significant differences were found in ADFI. Gilts (from 100 to 130 kg of BW) fed control diet ate 2.4% more feed (p < 0.001) and tended to grow 11% faster (p = 0.008) than gilts fed barley diet [22]. Other study [23] showed that the substitution of a conventional feed by a dietary barley reduced ADG and increased ADFI in gilts (from 45 to 92 kg of BW). On the other hand, previous study [24] did not find differences in ADG and ADFI between gilts (from 86 to 130 kg of BW) fed only granulated barley and those fed a control diet.

Line 172-173. There is no significant difference in the result of FCR. Thus, “80% wheat showed a slightly better feed conversion ratio” is an incorrect statement.

Response 17: Line 213-215: In the present experiment, no significant differences were noted between treatments with respect to ADFI and also FCR (FI was similar, but the fattening pig fed the diet containing 80% wheat showed only a trend of lower FCR).

Line 181: “by authors [14, 18], who replaced wheat with granulated barley in the diets.”. More details of the references [14,18] are needed to support your statement.

Response 18: Line 224-229: Similar results were reported by [22-24], who replaced wheat with granulated barley in the diets. Daza et al. [22] observed that the treatments had no effect on carcass traits except sirloin weight, which was lower in gilts fed barley diet than in those fed control diet. Similarly, Daza et al. [23] noted a lower loin weight in pigs fed barley than pigs fed conventional feed, but no differences were found for ham and foreleg weights or percentage in carcass. Also, Daza et al. [24] found similar results in heavy pigs fed mainly barley during growing-finishing period.

Line 211: more details of the reference of Nicolosi et al. [35] are needed to support your statement.

Ex. beta-glucan content, which animals, etc.

Response 19: Line 258-261:Because similar effect was noted in results of Nicolosi et al. [43] with application of β-glucan from yeasts. The yeast-derived β-glucan fiber significantly lowered total cholesterol concentrations and at the same time HDL-cholesterol concentrations rose in men.

Line 214: please revise “studies [20]” to “study [20]”

Response 20: Done

Line 215-217: more details of the reference [20] are needed to support the discussion. Ex. how much triticale fed to the pigs in the study of reference [20]. Also, you need to discuss the result of the present study considering the result of reference [20].

Response 21: Line 234-236: On the other hand, Turyk et al. [28] used concentrated feed with 80% triticale grain and reported leanness higher by 1% (p < 0.05) compared with a barley grain-based diet.

Line 262-266: Other relationships were observed in study [28] designed to check whether feeding of barley or triticale to fattening pigs (from 30.5 to about 110 kg of BW) influences serum lipid indices. The pigs fed more than 80% of barley containing in diets showed statistically non-significant increase in the level of total cholesterol and triglycerides but also in high-(HDL) and low-density lipoprotein (LDL) fractions.

Line 219-220: “The obtained results may indicate that lipid metabolism in the body depends on the energy source.” As per the nutritional values of the diets (Table 3), the energy value and crude fat content in the wheat 80% diet were higher than other diets. These differences may affect serum cholesterol components. I suggest you to openly discuss this difference of the diets.

Response 22: Line 268-279: The obtained results may indicate that lipid metabolism in the body depends on energy source. Reducing the level of energy and protein in the feed doses decreased the concentration of total cholesterol and LDL fraction [14]. Similar relationship was noted by Sawosz et al. [11], where were only a tendency towards decreasing the level of triglycerides, total cholesterol and LDL fraction in pigs fed low energy and protein diet. Unfortunately, also the concentration of HDL cholesterol was decreased in that group. Da Costa et al. [44] found that diets with reduced energy and protein contents increased potentials for substrate (protein, glycogen and lipid) turnover and mitochondrial function in growing pigs. Więcek et al. [14] did not find an unambiguous, statistically confirmed impact of the increase energy in the feed (by introducing the addition of 4% linseed oil to the diet) on the level of total cholesterol and its fractions. In the event of an excess of energy substrates in the body, then fatty acids are esterified and enter the blood in the form of very low-density lipoproteins [45].

Line 230: Blood components related to nitrogen metabolism (protein, albumin, globulin, urea) need to be discussed.

Response 23: Line 295-307: Animals fed with the reduced energy and protein level in feeding dose showed a tendency towards lower total protein, albumin and globulin content [14]. Our study confirmed that the level of total protein and albumin showed also a tendency towards decreasing when the value of hybrid barley in feed mixture increased. On the other hands, the addition of 4% linseed oil to the diet for fatteners increased the level of total protein and albumin [14].

Urea is a compound formed from the decomposition of proteins in the body, and it’s too high level in the blood serum indicates that animals do not use feed amino acids, which may be caused by too high protein content in the feed dose [53]. Reduction of protein content in the feed mixture (with appropriate amino acid balance) reduces the urea concentration in blood serum and decreases nitrogen excretion in urine, without deterioration of fattening results and carcass quality [54]. Study [55] showed that by lowering the protein level in the diet by 15%, a 35% reduction in serum concentration of urea can be achieved in fattening pigs. Other study [56] also reported reductions in urea values (5 to 8%).

Kind regards,

Anna Szuba-Trznadel

Reviewer 3 Report

The results of the study are interesting and give the basis for future studies but, honestly, I think that are very preliminary for a research article. I think that the manuscript would fit better as a communication. Even more when there are several data in the Results section that should be included in Material and methods because refer to the characteristics of the diet and not to the results obtained in the animals when applying the diets. The manuscript overall, having in mind the results obtained should be more assertive, less neutral and this would benefit from the structure of a communication rather than an extended research article. Abstract is poorly informative if read independently of the text. Increasing information will increase the interest of people looking at databases. Please, include relevant results in the abstract.

Author Response

Response to Reviewer 3 Comments

Comments and Suggestions for Authors

The results of the study are interesting and give the basis for future studies but, honestly, I think that are very preliminary for a research article. I think that the manuscript would fit better as a communication. Even more when there are several data in the Results section that should be included in Material and methods because refer to the characteristics of the diet and not to the results obtained in the animals when applying the diets. The manuscript overall, having in mind the results obtained should be more assertive, less neutral and this would benefit from the structure of a communication rather than an extended research article. Abstract is poorly informative if read independently of the text. Increasing information will increase the interest of people looking at databases. Please, include relevant results in the abstract.

Respone 1:

Thank you for your reviews.

I cannot agree with the opinion that the manuscript would fit better as a communication. In my opinion, the presented data are sufficient; some of them have been confirmed statistically, which seems interesting and gives the basis for further research.

The presented data, such as. components, were analyzed in the laboratory, therefore were showed in the Results. The data which was calculated (such as metabolizable energy) was included in Material and methods.

Results were improved, it means were more clearly presented.

The summary now contains more information (obtained results have been included in the abstract).

Kind regards,

Anna Szuba-Trznadel
